# Incremental improvements of 2030 targets insufficient to achieve the Paris Agreement goals

Andreas Geiges[1], Alexander Nauels[1,2], Paola Yanguas Parra[1], Marina Andrijevic[1,3], William Hare[1], Peter Pfleiderer[1,2,4], Michiel Schaeffer[1,5], and Carl-Friedrich Schleussner[1,2,4]

[1]Climate Analytics, 10961 Berlin, Germany
[3]IRITHESys, Humboldt University, 10117 Berlin, Germany
[2]Australian–German Climate and Energy College, University of Melbourne, Parkville, VIC 3010, Australia
[4]Potsdam Institute for Climate Impact Research, 14473 Potsdam, Germany
[5]Department of Environmental Sciences, Wageningen University and Research Centre, 6700 AA Wageningen, The Netherlands

**Correspondence:** Andreas Geiges (andreas.geiges@climateanalytics.org)

**Abstract.** Current global mitigation ambition up to 2030 under the Paris Agreement, reflected in the National Determined Contributions (NDCs), is insufficient to achieve the Agreement's 1.5°C long-term temperature limit. As governments are preparing new and updated NDCs for 2020, the question as to how much collective improvement is achieved is a pivotal one for the credibility of the international climate regime. The recent Special Report of the Intergovernmental Panel of Climate Change on Global Warming of 1.5°C has assessed a wide range of scenarios that achieve the 1.5°C limit. Those pathways are characterized by a substantial increase in near-term action and total greenhouse gas (GHG) emission levels about 50% lower than what is implied by current NDCs. Here we assess the outcomes of different scenarios of NDC updating that fall short of achieving this 1.5°C benchmark. We find that incremental improvements in reduction targets, even if achieved globally, are insufficient to align collective ambition with the goals of the Paris Agreement. We provide estimates for global mean temperature increase by 2100 for different incremental NDC-update scenarios and illustrate climate impacts under those median scenarios for extreme temperature, long-term sea-level rise and economic damages for the most vulnerable countries. Under the assumption of maintaining ambition as reflected in current NDCs up to 2100 and beyond, we project a reduction in the Gross Domestic Product (GDP) in tropical countries of around 60% compared to a no-climate change scenario and median long-term sea-level rise of close to 2m in 2300. About half of these impacts can be avoided by limiting warming to 1.5°C, or below. Scenarios of more incremental NDC improvements do not lead to comparable reductions in climate impacts. An increase in aggregated NDC ambition of big emitters by 33% in 2030 does not reduce presented climate impacts by more than about half compared to limiting warming to 1.5°C. Our results underscore that a transformational increase in 2030 ambition is required to achieve the goals of the Paris Agreement and avoid the worst impacts of climate change.

# 1  Introduction

Under the Paris Agreement of the United Nations Framework Convention on Climate Change (UNFCCC) governments have committed to holding temperature increase well below 2°C above pre-industrial levels and to pursue efforts to limit to 1.5°C (UNFCCC, 2015). However, current global efforts and targets are by far insufficient: aggregated mitigation targets under the Nationally Determined Contributions (NDCs), result in global warming of about 3°C (United Nations Environment Programme (UNEP), 2017; Climate Action Tracker, 2018; Climate Analytics, 2018).

The Special Report on Global Warming of 1.5°C (SR1.5) of the Intergovernmental Panel on Climate Change (IPCC) has emphasised the importance of near-term emission reductions to achieve the goals of the Paris Agreement (Masson-Delmotte et al., 2018). Pathways that achieve limiting warming to 1.5°C with no or limited overshoot require total greenhouse gas emission levels of 25-30 Gt $CO_2$eq/yr in 2030, about half of the 52-58 Gt $CO_2$eq/yr implied by current NDCs (Rogelj et al., 2018b). The IPCC further stressed that "rapid and far-reaching transitions" are required to achieve those emissions reductions and highlighted the importance of "fundamental societal and systems transitions and transformations" in helping to achieve the 1.5°C limit. In this context, it is important to emphasize that the scientific underpinning of the Paris Agreement temperature goal is linked to robust assessments of risks and impacts of climate change that would be avoided by achieving it (Schleussner et al., 2016b; Pfleiderer et al., 2018).

The IPCC Special Report further has provided comprehensive evidence on the impacts at global warming of 1.5°C and the impacts avoided compared to higher levels (Ove Hoegh-Guldberg et al., 2018). Particularly, substantially lower impacts are expected for extreme weather events (Seneviratne et al., 2018), water availability, regionally specific drought or flooding risks (Döll et al., 2018; Karnauskas et al., 2018; Hasson et al., 2019), crop production in particular in tropical regions (Faye et al., 2018; Schleussner et al., 2018b), circulation changes including extreme El Niño, persistence of weather patterns and tropical rainy season changes (Pfleiderer et al., 2019; Saeed et al., 2018; Wang et al., 2017), land and marine ecosystems (Warren et al., 2018; Schleussner et al., 2016a; Cheung et al., 2016), cryosphere changes including glacier and sea-ice loss (Laura and Dirk, 2018; Kraaijenbrink et al., 2017), (extreme) sea-level rise in particular beyond 2100 (Mengel et al., 2018; Schleussner et al., 2018a; Rasmussen et al., 2018), as well as economic damages (Burke et al., 2018; Pretis et al., 2018) and a wide range of other sectoral impacts (Arnell et al., 2018).

The findings of the IPCC were a key source of input into the Talanoa Dialogue process under the UNFCCC that has resulted in a "Call for Action" emphasising the need for increased near term ambition (UNFCCC, 2018). However, the window for strengthening the NDCs is closing quickly. By 2020, countries are required to come forward with new or updated NDCs over the time frame up to 2030 (UNFCCC, 2015).

Despite the scientific evidence for the need of profound increases in near-term emission reductions, it is far from certain that those materialise. Although governments may come forward with improvements of their commitments in their new or updated NDCs, those improvements fall short to deliver the emission reductions required on the global scale, but rather resemble gradual improvements to collective emission reductions efforts in 2030.

In the following, we explore different incremental global NDC update scenarios for 2030 and the implied global mean temperature increase up to the end of the century if a proportional level of mitigation effort is to continue throughout the century. We provide projections for selected climate impacts (sea-level rise , extreme temperaturesand economic damages) for current NDCs and gradual reduction pathways in comparison with a 1.5°C pathway. We are thereby linking near-term mitigation efforts directly to climate impact projections up to 2100 and beyond.

## 2 Methods

The analysis presented here combines a range of different approaches and methodologies ranging from an in-depth analysis of mitigation targets by big emitters to climate impact projections. Those will be elaborated upon in the following sections.

### 2.1 NDC pathways

The analysis of emission pathways builds on the methodology of the Climate Action Tracker (CAT) which estimates the collective result of current NDCs in global emissions. Under the assumption that the same global level of effort is kept after 2030, the CAT provided global mean temperature estimates of this pathway until the end of the century (Climate Action Tracker, 2015). Specifically, the CAT provides detailed assessments of pledges and policies of greenhouse gas (GHG) emission reductions by the Group of 20 (G20) plus a representative selection of minor emitters. Together the CAT countries[1] comprise about 80% of global emissions and 70% of total population (4.7 Billion in 2018). Although the list of CAT countries also includes a few minor emitters, the vast majority of the emissions from this group comes from the G20[2]. Therefore, in this analysis we use the combined emission reduction efforts derived from the CAT countries as a proxy for emissions reduction efforts by the political group of the G20. The extension to the global scale is then done following the assumptions about the emission trajectories by all other countries globally. For non-CAT countries, we assume that the emissions of these countries will either follow the countries Kyoto Protocol commitments (as applicable e.g. for Iceland) or a 'business-as-usual' (BAU) pathway. The BAU pathways used in this analysis are from the PRIMAP4 (Gütschow et al., 2016) baseline.

Based on this analysis, 2030 global emission levels based on the CAT assessment of current pledges (53 Gt $CO_2$eq, National emissions, excluding "Land-use, Land-use change and foresty" (LULUCF), international aviation and shipping emissions) and policies can be estimated. In order to relate the ambition reflected in the assessed NDCs in 2030 with the temperature goal of the Paris Agreement, an extension of emission scenarios until 2100 is required. This is done using the Constant quantile extension method (Gütschow et al., 2018) that is based on the assumption that the relative ambition level of climate policy is kept constant after the end of the NDC pathway. The extension is done using a database of emission scenarios from Integrated Assessment Models (IAMs) included in the 5th Assessment Report (AR5) (Clarke et al., 2014). The approach assumes that

---

[1]Argentina, Australia, Bhutan, Brazil, Canada, Chile, China, Costa Rica, Ethiopia, EU, Gambia, India, Indonesia, Japan, Kazakhstan, Mexico, Morocco, Nepal, New Zealand, Norway, Peru, Philippines, Russia, Saudi Arabia, Singapore, South Africa, South Korea, Switzerland, Turkey, United Arab Emirates, Ukraine, and the United States.

[2]Based on PRIMAP Hist data for national total GHG emissions (excluding land use and land-use chang). Emissions of the G20 countries sum up to 38.3Gt $CO_2$eq and for CAT countries to 40.3 Gt $CO_2$eq (Gütschow et al., 2016)

any emission target in a given year can be represented within the scenario space of IAM model results. This emission value in 2030 defines the implicit selection of IAM scenarios which are used to extend the pathway til the end of the 21$^{st}$ century, while maintaining the same level of ambition. This methodology ensures that the long-term projection is as consistent as possible with shorter-term action or pledges by accounting for the inertia of near-term actions.

## 2.2 Scenarios of incremental NDC improvements

Our "current NDC" ambition reference scenario corresponds to a modified version of the 2018 *CAT Pledges & Targets* pathway, which estimates global emissions levels implied by current NDCs. This CAT pathway accounts for all national emissions as aggregated Kyoto gas pathways excluding shipping, aviation and LULUCF. In order to obtain global emissions pathways, shipping emission are included based on ranges from RCP6.0, aviation emissions are based on data from (Owen et al., 2010) and LULUCF emissions are based on the median of baseline scenarios of land-use emissions from the LIMITS project (Kriegler et al., 2014). The 2018 CAT *Pledges & Targets* global emission pathway (excluding international bunkers and LULUCF) would lead a temperature increase of about 3°C in 2100.

To create a "current NDC" ambition reference scenario for our analysis of increased ambition beyond current NDCs, we modify the CAT scenario by assuming for all countries that they will reach the lowest emissions level implied by their current NDC targets (in case of multiple targets or a given range) or the projections of planned policies as estimated by the CAT (when these are lower than the country's NDC). Taking into account the fact that some countries are on track to overachieve their current NDC targets (e.g. India, Russia, Indonesia), as well as the conditionality of some NDC targets (e.g. The Philippines, Peru, Kazakhstan), 2030 total GHG emission levels implied by our reference scenario are 1.5 Gt $CO_2$eq lower than the 2018 *CAT Pledges & Targets* pathway (Climate Action Tracker 2018). Based on the pathway extension, our "current NDC" ambition reference scenario would lead to a median global warming of 2.8°C in 2100.

Starting from this "current NDC" ambition baseline that we define as the world's current highest ambition level, we create a number of NDC update scenarios, which are meant to represent different increments of improvement in ambition. Specifically, we assume a 5%, 10%, 25% and 33% reduction in global GHG emission levels below the "current NDC" reference pathway by 2030. We apply these reduction levels either to the CAT countries only (as the representatives of the largest emitters or G20 group) as well as to all countries globally. Comparing scenarios for big emitters and all countries will highlight the importance of the big emitters' reductions for the collective ambition reflected in the aggregated emission levels. In addition for consistency, the same reduction factors were applied to shipping, aviation and positive LULUCF emission for each scenario in 2030. We extend these incrementally strengthened NDC scenarios into pathways until 2100 following the constant quantile extension introduced above which is used for the reference scenario. This allows for an assessment of the implications of the gradual reductions for long-term temperature levels.

## 2.3 Deriving global mean temperature trajectories

The constructed GHG emission pathways (following AR4 global warming potentials) are then used to derive probabilistic projections for the global mean temperature (GMT) with the reduced complexity carbon cycle and climate model MAGICC6

(Meinshausen et al., 2011). MAGICC is an emulator which is calibrated against data from complex General Circulation Models (GCms), including climate sensitivity and carbon cycle information. Several studies have shown that model version 6 is able to capture both CMIP3 models(Meinshausen et al., 2011) and CMIP5 models (Rogelj et al., 2014; Nauels et al., 2017) responses well, also reflecting the climate sensitivity range assessed by IPCC AR5 (Rogelj et al., 2014) and the C4MIP carbon cycle response range (Friedlingstein et al., 2014). The MAGICC emulations reflect the complex model response ranges for the assessed scenarios in the calibration datasets, in particular the Representative Concentration Pathways (RCPs). Schwarber et al. (2019) showed that outside this calibration space, e.g., for the response under an instantaneous quadrupling of CO2, simplified models like MAGICC cannot reproduce the same level of complexity in their response as comprehensive climate models. However, since the presented analysis of NDC pathways is well within the range of the scenarios used for the calibration, we consider MAGICC6 a suitable tool for our analysis and projections of GMT responces. In this context it is worth noting that overarching efforts to reconcile observations with complex model responses have been successful as well (Cowtan et al., 2015; Hausfather et al., 2020). Numerous studies have been able to show that CMIP models can be reconciled well with observations if the right methodologies are applied (Cowtan et al., 2015; Tokarska et al., 2019; Hausfather et al., 2020). To allow for the comparison with a representative 1.5C consistent pathway, MAGICC6 is also forced with SSP1-RCP1.9 type emissions (Rogelj et al., 2018a), normalized to the year 2010 emissions of the CAT pathways to ensure a consistent experimental setup. The MAGICC model is run from 1750 to 2300 to also shed light on longer-term impacts like global mean sea-level rise. It is important to stress that for this longer-term outlook, radiative forcing levels for the $22^{nd}$ and $23^{rd}$ centuries are held constant at 2100 levels for all pathways, making the post-2100 model responses more stylized than for the $21^{st}$ century.

## 2.4  Climate Impacts

In order to illustrate the implications of different temperature trajectories, we extend our analysis with additional climate impacts for three selected pathways: the current NDC pathway as reference (**NDC**), a selected gradual improvement scenario for the big emitters (CAT countries) with a reduction in GHG emissions by 33% (**BE33**) and a 1.5°C scenario based on SSP1-RCP1.9 (Rogelj et al., 2018b) (**1.5°C**).

Global mean temperature is an established metric that allows to approximate a range of different climate impacts. However, regional or sectoral changes can be much more pronounced than those in global mean temperature. As established in the IPCC SR1.5, a wide range of vulnerable systems are sensitive to temperature differences of 0.5°C or less, including terrestrial and marine ecosystems such as coral reefs, cryosphere changes and extreme weather indices (Schleussner et al., 2016b).

Providing a comprehensive analysis of the differences in climate impacts would go beyond the scope of this analysis that focuses on the implications of different 2030 NDC update scenarios. However, to illustrate the relevance of these differences for a range of different climate impacts, we have chosen three exemplary impact indicators: a time-lagged response (long-term sea level rise), an extreme event indicator (extreme hot days), and economic damages.

### 2.4.1 Long-term sea-level rise

Global mean sea-level rise (GMSLR) projections are generated with the MAGICC sea-level model (Nauels et al., 2017). For the period 1850 to 2300, the model emulates IPCC AR5 consistent process-based model projections for thermal expansion, glacier mass loss, Greenland and Antarctic ice sheet contributions, and also includes a land water storage estimate which is independent from the climate change signal. While the Antarctic sea-level component accounts for rapid dynamics captured by (Levermann et al., 2014), it does not reflect the proposed process of marine ice cliff instability (Deconto and Pollard, 2016), that could increase sea-level estimates for high emission scenarios but is scientifically debated still (Edwards et al., 2019; Golledge et al., 2019). To this end, the provided sea-level projections can be interpreted as conservative estimates for the longer-term sea-level response.

### 2.4.2 Extreme temperature

We present changes in hot extremes as land fraction distributions of changes in the intensity of the hottest day in a year (TXx), following the method introduced by Fischer et al. (2013). This analysis is based on a time-slicing approach: for each model, a 21-year time period is selected for which the averaged global warming corresponds to the end-of-century GMT increase given by the respective scenarios (see Table 1). At each grid-cell, the hottest days of a year (TXx) are averaged over the selected 21-year period and these averages of all selected CMIP5 runs are merged into one TXx change distribution per scenario and region. Weighting the grid-cells by their area, a land fraction distribution is calculated for each region (as shown in Fig. 2). For each model we selected a run in the lowest emission scenario for which the desired GMT value is reached. Doing so, we ensure that the warming during a 21-year period is minimal and that all models have an equal contribution to the result. Exact numbers for all regions can be found in the suplementary material (see below).

### 2.4.3 Economic damages

Expressed in terms of the gross domestic product (GDP), economic damages are calculated based on the methodology of Burke et al. (2018). The method combines the estimates obtained from the historical relationship between GDP growth and temperature variability, and projected future temperature changes. With our own temperature pathways, GDP is estimated at the degrees of warming reached with the current NDCs, BE33, and a 1.5°C pathway, and then compared to the GDP in a counterfactual scenario without climate change. The impacts are calculated for mid-century (2046-2065) and end-of-century (2081 - 2100).

To downscale GMT differences to gridded change patterns in annual mean temperature, we calculate downscaling factors for each model based on differences in annual mean temperature between two 20-year periods (2046-2065 and 2081-2100) and a reference period (1986-2005) for the three scenarios. Then we apply the median downscaling factor of each model (see Figure S1) to the GMT differences of the selected scenarios to get local changes in GMT.

| Scenario | Big emitters (BE) | | All countries (ALL) | |
|---|---|---|---|---|
| | Emissions 2030 [$CO_2$eq] | GMT 2100 | Emissions 2030 [$CO_2$eq] | GMT 2100 |
| High CAT Pledges & Targets pathway (as reference) | - | - | 56.2 Gt | 3.0 °C |
| NDC reference scenario (this study) | - | - | 54.8 Gt | 2.8 °C |
| NDC 5% emission reduction | 52.7 Gt | 2.56 °C | 52.2 Gt | 2.52 °C |
| NDC 10% emission reduction | 50.6 Gt | 2.43 °C | 49.6 Gt | 2.35 °C |
| NDC 25% emission reduction | 44.3 Gt | 2.1 °C | 41.8 Gt | 2.0 °C |
| NDC 33% emission reduction | 41.0 Gt | 1.9 °C | 37.75 Gt | 1.75 °C |

**Table 1.** Overview of the evaluated scenarios, including details on projected global greenhouse gas emissions for the year 2030 and the projected median increase in Global Mean Temperature (GMT) for the year 2100 relative to 1750. The emission reduction levels in 2030 are applied to either only big emitters (BE) or all countries (ALL)

## 3   Results

The ambition reflected in current NDCs as assessed in the 2018 *CAT Pledges & Targets* pathway would put us on track for about 3°C of temperature increase in 2100 (see Table 1). Applying a consistent model setup, our slightly more optimistic "current NDC" reference scenarios used here would lead to a warming reduction of about 0.2°C, implied by a reduction in 2030 GHG emission levels of about 1.5 Gt $CO_2$eq compared to the 2018 CAT pathway.

None of the increased ambition pathways considered here limit median warming to 1.5°C by 2100, which would require a reduction in 2030 NDC GHG levels of about 50% (Masson-Delmotte et al., 2018). Only the 25% and 33% reduction scenarios for all countries limit median warming to under 2°C (see Table 1 and Figure 1). A median estimate of less than 2°C does however not imply that such pathways are in line with the "well below 2°C" limit set out in the long-term temperature goal of the Paris Agreement (Schleussner et al., 2016b). Smaller incremental improvements of a 5 or 10% increase in ambition would not bring median estimates for GMT increases close to 2°C.

Scenarios in which emission reductions are limited to big emitters only lead to higher warming levels. The temperature outcomes for the smallest ambition increase in the scenarios studied (5% and 10%) are very close. This indicates that the impact of a NDCs ambition increase by small emitters is very limited at the global level if bigger emitters do not increase ambition at least proportionally. For larger increases in NDC ambition (25% and 33%), the difference in implied 2100 GMT increase between the big emitters and all country scenario increases (see Table 1). Conditional on big emitters leading the way, the importance of the contribution of relatively small emitters thereby increases under scenarios of more ambitious global ambition. This result demonstrates that getting close to the Paris Agreement's long-term temperature goal will require comparable levels of action by all emitters, not just the largest.

### 3.1 Long-term sea-level rise

2100 GMSLR projections under the NDC reference scenario yield a median of around 64 cm (66% model range: 50 to 81 cm) (see Table 2 and Figure 2). If major emitters increased their NDC ambition by 33% (BE33), 2100 GMSLR would be around 10 cm lower, namely 54 cm (43 to 68 cm). The scenario of a 1.5°C consistent ambition level shows a projected GMSLR of 45 cm (36 to 57 cm) in 2100 which is an additional reduction of around 10 cm. When looking beyond 2100, the sea-level rise implications of the selected scenarios become more pronounced. For 2300, the stylised pathway extensions yield around 190 cm (140 to 250 cm) of GMSLR under the NDC reference scenario, 140 cm (110 to 180 cm) for the BE33 case, and around 100 cm (80 to 130 cm) for the 1.5°C consistent pathway.

### 3.2 Extreme temperature

Figure 3 shows the changes in the hottest day in a year (TXx) with respect to 1986-2005 levels for each scenario globally (lower left panel) and for regions used in IPCC reports. In all regions, the differences between no increase in ambition, a 33% increase in ambition and a 1.5°C scenario are clearly distinguishable. Under a 1.5°C scenario (around 0.9°C above the 1986-2005 level), for 50% of the land area TXx would increase by at least 1.1°C (compared to current conditions). Only a few places would experience increases as high as 3°C in TXx. For the BE33 scenario, half of the land area would experience an increase in TXx of at least 1.8°C. In this scenario, for 39% of the land area a 2°C increase in TXx is projected whereas in the 1.5°C scenario this increases is only projected for 11% of the land area.

Under the NDC scenario, changes in TXx would be most pronounced. The global median increase of TXx under this scenario is projected to be 2.7°C above the 1986-2005 level. The increase in the high-end tail is most pronounced under this scenario with 10% of the land area experiencing increases in TXx of over 4°C above the 1986-2005 levels.

### 3.3 Economic damages

In line with previous assessments (Diffenbaugh and Burke, 2019; Burke et al., 2018), most countries and in particular those in tropical regions are projected to experience economic damages from temperature increase under all emission scenarios. Note that the model by Burke et al. (2018)) does not consider the effects of sea-level rise, extreme weather events, or other non-linear trends that would likely exacerbate the current estimates. Figure 4 shows economic damages for selected vulnerable regions, grouped in four categories based on their geographical region or level of development: Latin America, South Asia, Least Developed Countries (LDCs) and Small Island Developing States (SIDS).

Compared to a no climate change scenario, thr estimated median reduction in GDP per capita around 2050 for the four country groups ranges between 11% and 14% for the 1.5°C pathway, 14% and 17% for the BE33 scenario, and between 17% and 20% reduction for the baseline pathway of the current NDCs. Variations between the country groups are small, though they are consistently highest for the LDCs. The differences between the scenarios increase further towards the end of the projection period. In 2100, the median projection of damages to GDP per capita range between 29% and 34% for the 1.5°C pathway, 43% and 50% for the 33% increase in NDCs, and between 58% and 66% reduction for no change from the current NDCs,

| Climate Impact | Year | NDC reference scenario | 33% ambition increase for big emitters | 1.5°C scenario |
|---|---|---|---|---|
| Median global mean sea-level rise relative to 1986-2005 | 2100 | 64cm | 54cm | 45cm |
| | 2300 | 190cm | 140 cm | 100cm |
| Median increase in annual maximum temperature (TXx) relative to 1986-2005 | 2100 | +2.7°C | +1.8°C | +1.1°C |
| Median GDP reductions for LDC countries relative to a no-climate change scenario | 2050 | -20% | -17% | -14% |
| | 2100 | -66% | -50% | -34% |

**Table 2.** Selected climate impacts under different scenarios of mitigation ambition: Global mean sea-level rise, annual maximum temperature (TXx) increase, GDP reductions for Least Developed Countries (LDCs). For more quantitative information on uncertainties please see figure 2 to 4.

indicating profound economic risks for developing countries under scenarios of current ambition. Median economic impacts can be halved by achieving a 1.5°C scenario.

## 4 Discussion and conclusions

We have provided a detailed analysis of the implied consequences of present day NDC ambition levels as well as incremental improvement scenarios up to a 33% reduction relative to present day NDCs emission levels in 2030. In line with the 1.5°C Special Report by Masson-Delmotte et al. (2018), we find that such gradual improvements are insufficient to achieve the 1.5°C limit, which would require 2030 GHG emission levels to be about halved compared to current NDCs. As expected, more ambitious mitigation targets for big emitters have the biggest effect on GMT and related impacts. Therefore, scenarios for big emitters only show relatively similar temperature responses for high GMT trajectories compared to scenarios for all countries. However, the increasing differences in temperature outcomes between these two scenario groups show that, with increasing ambition, the relative importance of the contributions from small emitters grows (compare Table 1). This has important implications for climate policy as it underscores that big emitters need to spearhead global efforts, but that in order to achieve the goals of the Paris Agreement no country can stay behind and all are needed to improve collective ambition sufficiently to limit warming to 1.5°C.

Our estimates of selected climate impacts relating to current NDCs point to substantial additional impacts implied by current trajectories. Even if big emitters increase their ambition level by a third, this will only reduce about half the inferred impacts including for sea level rise, extreme temperatures and economic damages compared to what can be achieved by limiting warming to 1.5°C.

The consequences for affected population around the globe and specifically vulnerable regions such as least developed countries or Small Island States will be profound (Schleussner et al., 2018a). Our findings are in line with other studies reporting substantial impact reduction potential at 1.5°C compared to higher levels of warming (Arnell et al., 2018; Schleussner

et al., 2016a; Byers et al., 2018). The implications of a lack of ambition towards achieving beyond gradual improvements will therefore manifest in a broad range of impacts beyond the limited set of impacts studied here. The IPCC Special Report on 1.5°C has identified a range of key reasons for concern as high above 1.5°C including extreme weather events, unique and threatened systems as well as globally unequal impacts (Ove Hoegh-Guldberg et al., 2018). Our results confirm these findings and add additional information also in comparison with trajectories implied by current NDCs. Differences in median sea level rise between a 1.5°C and NDC scenario in 2100 amount to about 20cm, as much as the world has experienced over the observational period, which has already contributed significantly to the occurrence of coastal flooding (IPCC, 2019). Beyond 2100, the difference could amount to almost one meter or more until 2300 (compare Table 2). For extreme heat, we find a doubling in impacts between 1.5°C and NDC pathways (compare Figure 3), with profound regional differences. For Central Europe, for example, we project that about 40% of the land-area would experience a TXx increase under current NDC pathways of about 4°C. For tropical regions, exceeding a new climate regime in terms of extreme temperatures will be reached already for 2°C warming (Russo et al., 2016). Exceeding 1.5°C will also substantially increase the risks of exceeding tipping points of the earth system (Schellnhuber, H. J. Rahmstorf and Winkelmann, 2016).

Our analysis does not aim at deriving conclusions about the global impact of *individual* proposed NDC updates, but rather about the overall *collective aggregate ambition* increase that is needed in the short term to keep the door open for ambitious emissions reductions in the long-term. This means that the reduction numbers cannot be to easily extrapolated to individual NDCs of countries. First and foremost, actual domestic emissions reductions will differ strongly among countries, depending on technically and economically feasible reduction potentials. In addition, the additional emissions reductions to be achieved elsewhere via contributions to e.g. international climate finance, as well as international finance received by countries, may depend on assumptions of fair share and equity(see e.g. Robiou Du Pont et al. (2017)) that need to be considered carefully .

Individual NDCs may have very different types of targets, with mixed coverage of sectors and different levels of uncertainty around the emission levels implied by the targets included in the NDCs. The overall percentage reductions presented here therefore need to be translated back to each country's specific NDC "language" to understand how the NDC update announcements compare to the different levels of ambition described. For countries where current policies indicate an overachievement of their NDC targets, the percentage improvement presented here refers to emissions levels in line with these current policy projections (see above) and thus translates to a higher percentage improvement from the current NDC target.

Finally, our "current NDC" baseline pathway refers to emissions excluding LULUCF activities. NDC contributions to LULUCF pathway are assumed to drop by the same percentage, thus we are assuming that the contribution to the NDC enhancement will be equal for LULUCF and no-LULUCF sectors. It is to be expected, however, that for a large number of countries the LULUCF sector contributions to a more ambitious NDC would be more than proportionally with respect to non-LULUCF sector. These considerations should be carefully examined when judging the ambition improvements of individual NDCs, in particular given issues regarding transparency and ambiguity of the treatment of LULUCF in the current NDCs. The ambiguous land use mitigation targets, provided by most countries results in an uncertainty of about ~3 GtCO$_2$/year in global land use emissions in 2030 (Fyson and Jeffery, 2019) .

Our results show the need for a transformational increase in 2030 ambition by countries to achieve the 1.5°C limit and to avoid the impacts of exceeding this level of warming. While it is necessary – and essential – that these ambition increases need to be spearheaded by the big emitters, it also clear that this is not sufficient and hence all countries need to contribute their fair share reflected in their 2020 NDCs in order to achieve the transformational change in near term ambition required to meet the Paris Agreement's long term temperature goal.

*Code and data availability.* All data and code for visualizing the results of the study can be found here: https://gitlab.com/ageiges/ndc_ambition_esd_paper_material

*Author contributions.* AG, PP, CFS, AN, MS and WH designed the scenarios, AG computed the emission pathways, AN and AG provided the temperature projections. AN provided the sea-level projections, PP the temperature extremes and MA the economic damages. CFS lead the writing of the manuscript with contributions of all authors.

*Competing interests.* The authors declare that they have no conflict of interest

*Acknowledgements.* We acknowledge the modelling groups, the Program for Climate Model Diagnosis and Intercomparison, and the WCRP's Working Group on Coupled Modelling for their roles in making available the CMIP multi-model datasets. Support for this dataset is provided by the Office of Science, US Department of Energy. We thank Marshall Burke for making his code underlying our economic analysis openly available on GitHub. AG, PP, MS, WH acknowledge support by the German Federal Ministry for the Environment, Nature Conservation and Nuclear Safety (16_II_148_Global_A_IMPACT). MA, CFS and PP acknowledge support by the German Federal Ministry of Education and Research (01LN1711A). AN is supported by the European Union's Horizon 2020 research and innovation programme under Grant Agreement N°820829.

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

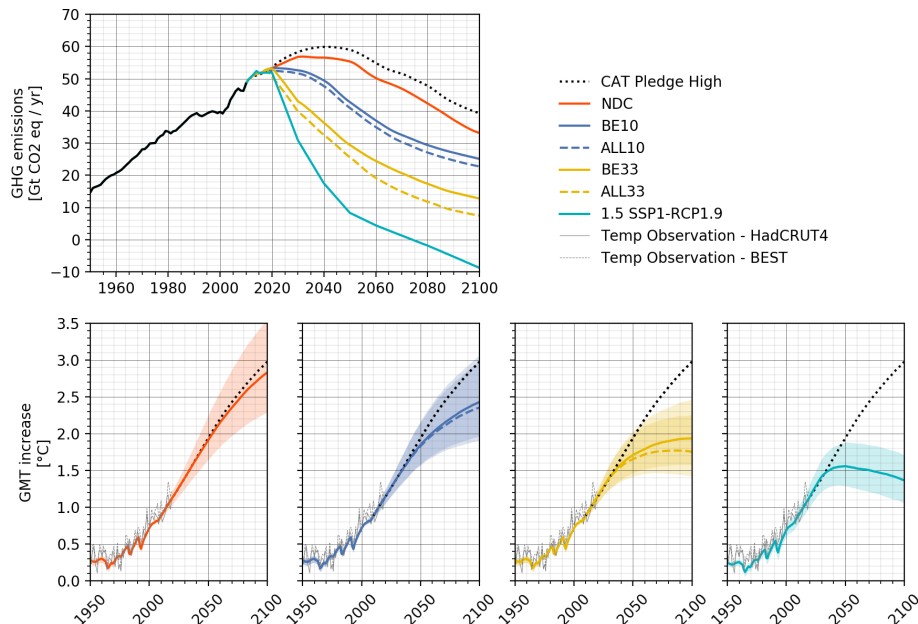

**Figure 1.** Historical and annual global Greenhouse Gas (GHG) emission and Global Mean Temperature (GMT) trajectories for the NDC reference scenario and two NDC improvement scenarios with a 10% and 33% reduction of 2030 emission levels relative to the NDC reference. Scenarios for Big Emitters are labeled "BE" and scenarios for all countries using "ALL". Also shown is the observed global temperature change based on the HadCRUT4 (solid grey)4 and BEST (dashed grey) datasets (REF). All GMT estimates are provided relative to the 1850-1900 average. Annual global GHG emissions are provided in gigatonnes $CO_2$ equivalent (Gt $CO_2eq$).

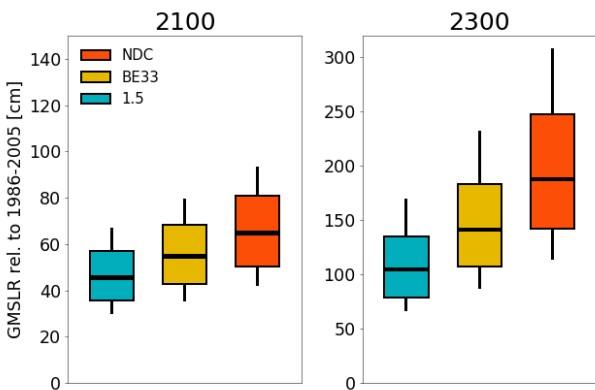

**Figure 2.** Global mean sea-level rise (GMSLR) projections under the NDC reference scenario (NDC), the CAT-based scenario for a 33% increase in NDC ambition of big emitters (BE33), and a scenario consistent with limiting 21st century warming to 1.5 C (1.5 C) for the years 2100 and 2300, relative to the IPCC AR5 reference period 1986-2005 in centimetres. Median values (thin horizontal lines) are provided together with the 66% model ranges (boxes) and 95% model ranges (whiskers). Please note the different y-axis scales.

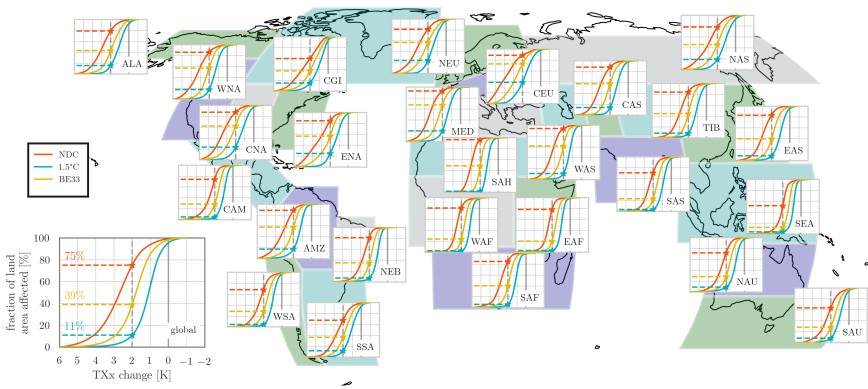

**Figure 3.** Changes in hot extremes (TXx) in the period 2081-2100 relative to 1986-2006 for the NDC scenario (red), the BE33 scenario (yellow) and the 1.5°C scenario (cyan). Changes are presented as land area fraction (y-axis) affected by changes in TXx (x-axis) for 26 world regions and globally. Cumulative distributions are based on an area weighted aggregation of all TXx change values projected at grid-cells within a region and across climate models. A vertical dashed gray line indicates TXx changes of 2°C. The asterixis and the horizontal lines going from the 2°C line to the left y-axis show the land fraction that is affected by an increase in TXx of at least 2°C.

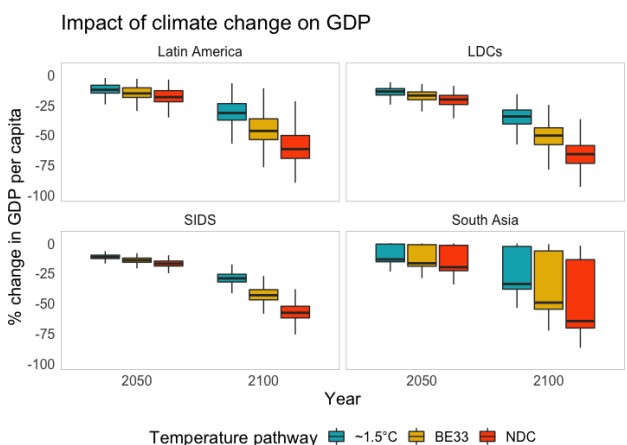

**Figure 4.** Economic damages under different scenarios of GMT increase. The boxplots contain estimates for different GCMs, and show the percentage difference between GDP per capita under selected temperature pathways (no change in current NDCs, BE33 scenario resembling a 33% change in NDCs of big emitters and a 1.5°C pathway) and GDP per capita under a no climate change scenario. The lower and upper hinges correspond to the first and third quartiles (the 25th and 75th percentiles). The upper whisker extends from the hinge to the highest value that is within 1.5 the interquartile range of the hinge. Estimates are given for mid-century (2046-2065) and end-of-century (2081-2100). Countries are grouped by either geographical regions (South Asia and Latin America) or political groupings following the UN classifications (Small Island Development States, SIDS and Least Developed Countries, LDCs).