# Peer review of "Incremental improvements of 2030 targets insufficient to achieve the Paris Agreement goals"

_Earth System Dynamics, 2019_

## Referee Comment (RC1) · Anonymous Referee #1 · 22 Dec 2019

**1. Summary and general comments**

This paper examines GHG emissions out to 2030 as pledged in the NDCs of the Paris Climate Agreement and create scenarios for further reductions, then extends these emissions scenarios to 2100 with statistically analogous scenarios from an existing database. The paper translates these emissions scenarios to end-of-century global mean temperature anomalies (relative to preindustrial) using a simple climate model, MAGICC6, and examines the consequences of such temperature increases for sea level rise, maximum temperature days, and economic damages. Based on this analysis, the paper states that the global community needs to reduce emissions noticeably

more than they have already committed to do with the existing NDCs if they wish to avoid climate catastrophe, as none of the scenarios presented herein provide end-of-century temperature below the Paris Climate Agreement goal of 1.5 degrees of warming.

Overall, this paper is very well-written in terms on language and accessibility, with very few typographical or grammatical errors as well as a relatively straightforward and clear writing style. The paper is informative without being overly technical, and does a great job of succinctly placing the more technical modeling results in the context of a range of real-world consequences.

However, there is one significant problem with the submitted manuscript that should be addressed prior to acceptance for publication. As detailed below, the MAGICC6 model tends to warm more quickly than the aggregate of CMIP5 atmosphere-ocean global climate models. Furthermore, the CMIP5 AOGCMs warm more quickly than observations. As a result, the paper as submitted provides an overly pessimistic view of the GHG emission reductions that will be need to meet the Paris Climate Agreement. Upon revision, this tendency for MAGICC6 to overestimate warming needs to be addressed.

2. Specific comments

The relationship between 2030 emissions and 2100 temperatures constitutes the core of this paper. The methods state that MAGICC6 is run with the climate sensitivity range of AR5, largely driven by CMIP5, and the carbon cycle range of C4MIP. However, MAGICC6 tends to exhibit a faster increase in global mean surface temperature (GMST) than AOGCMs when run with comparable values for climate sensitivity (Schwarber et al. 2019, ESD, doi: 10.5194/esd-10-729-2019 – particularly figure 4). Also, the CMIP5 AOGCMs tend to warm more rapidly than observed GMST (AR5 figure 11.25; Fyfe, Gillett, and Zweirs 2013, Nature Clim Change, doi:10.1038/nclimate1972; Millar et al. 2017, Nature Geoscience, doi:10.1038/ngeo3031). Projections of future GMST provided by observationally constrained models are similarly noticeably

lower than those from free-running AOGCMs (Chylek et al. 2016, Climate Dynamics, doi:10.1007/s00382-016-3025-7; Salawitch et al. 2017, Springer International Publishing, doi:10.1007/978-3-319-46939-3).

While this paper is in line with IPCC's estimate of necessary reductions from the 1.5 degree special report, the report largely relies on just MAGICC6 and FAIR (another simple climate model) to make these emissions statements. FAIR is only marginally better than MAGICC6 at meeting the median CMIP5 temperature response, as shown in Schwarber et al. 2019. MAGICC6 has enough tunable parameters to produce results in-line with CMIP5 models (Hartin et al. 2015, GMD, doi:10.5194/gmd-8-939-2015; Meinhausen et al. 2011b, ACP,* doi:10.5194/acp-11-1457-2011) and, presumably, observed GMST. However, these scenarios are not explored, and represent a shortcoming in the submitted paper. As a result, the warming scenarios used in this paper likely have a noticeable hot bias.

*(Meinhausen et al. 2011b, ACP, the companion paper to the one cited; it shows comparisons of MAGICC6 being both in line with and warmer than AOGCM results under different presentations)

The consideration above would, of course, bring into question the core proclamation that none of the suggested NDC improvement scenarios meet the 1.5 degree goal. If the actual climate system does not warm as much under the ALL33 scenario (or any other scenario) as suggested by the MAGICC6 model runs, the likelihood of meeting the 1.5 degree goal would be much higher than suggested in the submitted paper. This criticism is not to suggest that the core message will change dramatically if the potential hot bias is accounted for – it could still very well be that the ALL33 scenario does not end below the 1.5 degree goal, or does end below 1.5 degrees but not strongly enough to say it would do so with sufficiently high confidence. However, for this paper to be considered for publication, more justification (and/or context) needs to be given for the amount of warming that results from each emissions scenario, with careful attention to how well MAGICC6 can simulate observed warming.

On one hand, if this particular configuration of MAGICC6 does definitively have a hot bias (as indicated by papers cited above), its use could be justified by stating that application of a hot model means any emissions scenario that meets the 1.5 degree goal under this analysis should meet that temperature goal no matter what model (or ensemble of models) is considered. That is, intentional underestimation in an emission scenario's potential success has the policy benefit of higher confidence in that scenario overall. On the other hand, statements such as the penultimate sentence of the abstract (lines 15-17), the opening paragraph of section three (lines 156-159), or the "clear evidence" conclusion (line 257) should probably be made based on likely warming scenarios, not on warmer-than-average scenarios.

3. Technical/other comments

Line 112 – spacing

Lines 136 to 140 – sentences are a little hard to follow

Line 165 – verb confusion

Lines 173 to 176 – I don't follow this sentence at all

Line 180 – the unit "cm" is separate from its numbers on the previous line

Line 181 to 182 – the two parts of the sentence are ordered backwards from what a reader might normally expect

Table 2, NDC column, 2300 row – spacing

Lines 219 to 220 – the last phrase here feels awkwardly worded

Line 245 – unnecessary comma after first word; also "...each country's specific..." (possessive)

Line 249 – unnecessary capitalization ("Land use")

Overall – aside from the major reservation about warming bias in the chosen model,

this is a very well-written and well-reasoned paper

---

## Referee Comment (RC2) · Anonymous Referee #2 · 28 Jan 2020

In this paper, the authors first estimate emissions related to Nationally Determined Contributions scenarios, and the calculate corresponding climate impacts. The paper is well written albeit brief in the analysis. The study design is particularly interesting and novel and the methods used are comprehensive. Presentation of the climate impacts and results in general, could be improved. The results section is shorter than the methods. Discussion could be made more interesting. Overall, this will be a very good paper, but I think there are some changes that are within the authors reach to greatly improve the work. Criticisms below.

The authors appear to have made no efforts to make neither the results nor code avail-

[Figure]

able. I urge the authors to make, at least the additional data used to make the figures available. Additionally, data on the NDC emissions scenarios and corresponding temperature outcomes from MAGICC would also be useful to the community (e.g. Fig 1).

One weakness, given the substantial work already done, is why not more climate impacts considered. The authors have gone great efforts in the first half of the work relating to NDCs, then only present three first order climate impacts. In the paper the authors estimate sea level rise impacts, extreme temperatures and economic damages – from different models/approaches. Considering Precipitation, both high and low indicators, would have surely been straight forward to add to the statistics on extreme temperatures, for example.

Figure 4 is nicely designed, essentially useless. Even if one could accurately read such small graphs, knowing the change in probability density function conveys very little information. You have made great efforts to make the first half of the paper policy-relevant – linking to NDCs etc – and then the way the Temperature and SLR impacts are presented and discussed is without value. At least CDFs could be used to show % of land impacted, or population impacted. In any case the figures are so small they cannot be used – the only point is to demonstrate that you have the information but have no intention for other people to use it.

Perhaps an illustrative diagram in section 2.1 would help describe the methods. I'm not convinced by using Txx – hottest day in the year, because I am unconvinced that the GCMs are able to consistently, especially across regions, reproduce the single hottest day each year. Why pick such a difficult target, when perhaps 5th hottest day each year, or a p99 over the 21-year period would probably equally be sufficient to estimate the changing temperature distribution and is likely much better reproduced by the models? If you insist on this indicator, can you provide some validation that at least the GCMs used accurately reproduce Txx for the historical period with error less than 0.6 degC – because that's the difference between your BE33 and 1.5C scenarios.

Figure 3 – why is 1.5 in red, and NDC in Blue? This doesn't make sense and is opposite to Fig 4 and Fig 2 colour schemes. Are there no uncertainty ranges associated with the economic impacts assessment from the Burke methodology?

It's not clear what is the point of the including these climate impacts. The results are presented in a mechanical fashion. Why they are included, and justification for the specific impacts, is not provided. In the discussion there are only 4 lines about them. Try to explain the "so what" for the reader. What does 1m sea level rise mean? Does 1-3degC really make a difference for the hottest day of the year. It doesn't sound like much to regular people, but if you're an expert, we know it has impacts on animals, vulnerable people, chance of crop failure, labour productivity, power plant efficiencies, peak electricity demands, rail tracks and roads – and on and on and on. So try and bring this perspective to the reader, on why 1-3 degC change in hottest day of the year is significant. The same applies to sea level rise that is in the order of centimeters... doesn't sound like much, but if you live in Netherlands, or Bangladesh, or Miami – its terrible news!

Supplementary information is quite concise and no results provided in data form.

Line 79: Spell out AR5

Line 198-199 – you say 4degC here – but you should clarify that this is for the hottest day only, not mean temperature rise, as it could easily be misunderstood.

Line 231: You say results in line with other studies, yet you only provide one citation? Consider works that could back up this statement e.g. by Piontek et al (2014, PNAS), Scheussner et al (2016, ESD), Byers et al (2018, ERL), Mora et al (2018, Nat CC)

---

## Author Comment (AC1) · 9 Mar 2020

Geiges et al.: "Incremental improvements of 2030 targets insufficient to achieve the Paris Agreement goals" https://doi.org/10.5194/esd-2019-54 Compiled author responses to reviewer 1 comments

REVIEWER #1

General Comment

Reviewer Comment (RC1.00): This paper examines GHG emissions out to 2030 as pledged in the NDCs of the ParisClimate Agreement and creates scenarios for further

reductions, then extends these emissions scenarios to 2100 with statistically analogous scenarios from an existing database. The paper translates these emissions scenarios to end-of-century global mean temperature anomalies (relative to preindustrial) using a simple climate model, MAGICC6, and examines the consequences of such temperature increases for sea level rise, maximum temperature days, and economic damages. Based on this analysis, the paper states that the global community needs to reduce emissions noticeably more than they have already committed to do with the existing NDCs if they wish to avoid climate catastrophe, as none of the scenarios presented herein provide end-of-century temperature below the Paris Climate Agreement goal of 1.5 degrees of warming. Overall, this paper is very well-written in terms of language and accessibility, with very few typographical or grammatical errors as well as a relatively straightforward and clear writing style. The paper is informative without being overly technical, and does a great job of succinctly placing the more technical modeling results in the context of a range of real-world consequences. However, there is one significant problem with the submitted manuscript that should be addressed prior to acceptance for publication. As detailed below, the MAGICC6 model tends to warm more quickly than the aggregate of CMIP5 atmosphere-ocean global climate models. Furthermore, the CMIP5 AOGCMs warm more quickly than observations. As a result, the paper as submitted provides an overly pessimistic view of theGHG emission reductions that will be needed to meet the Paris Climate Agreement. Upon revision, this tendency for MAGICC6 to overestimate warming needs to be addressed.

Author Response (AR1.00): We would like to thank the reviewer very much for this thorough and constructive review and are pleased that our work is considered to be informative and valuable. We hope that our below responses will address the main reviewer concern that MAGICC6 would be warming too quickly, potentially leading to an overestimation of required emission reduction needs. All responses to the reviewer comments are listed below.

Major Comments

RC1.01: The relationship between 2030 emissions and 2100 temperatures constitutes the core of this paper. The methods state that MAGICC6 is run with the climate sensitivity range of AR5, largely driven by CMIP5, and the carbon cycle range of C4MIP. However, MAGICC6 tends to exhibit a faster increase in global mean surface temperature (GMST) than AOGCMs when run with comparable values for climate sensitivity (Schwarber et al. 2019, ESD, doi: 10.5194/esd-10-729-2019 – particularly figure 4). Also, the CMIP5 AOGCMs tend to warm more rapidly than observed GMST (AR5 figure11.25; Fyfe, Gillett, and Zwiers 2013, Nature Clim Change, doi:10.1038/nclimate1972; Millar et al. 2017, Nature Geoscience, doi:10.1038/ngeo3031). Projections of future GMST provided by observationally constrained models are similarly noticeably lower than those from free-running AOGCMs (Chylek et al. 2016, Climate Dynamics,doi:10.1007/s00382-016-3025-7; Salawitch et al. 2017, Springer International Publish-ing, doi:10.1007/978-3-319-46939-3).

AR1.01: We would like to thank the reviewer for raising these two crucial issues: MAG-ICC6 vs AOGCM/ESM warming rates and AOGCM/ESM warming vs observations. We very much appreciate the opportunity to clarify these issues by providing further information and evidence.

MAGICC6 vs AOGCM/ESM warming: MAGICC is an emulator which is calibrated against AOGCM data including climate sensitivity and carbon cycle information. By definition, MAGICC is designed to capture the magnitude and rate of change from complex models and it has been shown in numerous publications that model version 6 is able to do so for both CMIP3 models (Meinshausen et al 2011, Figure 2) and CMIP5 models (IPCC AR5 WGI, Figure 12.36, Nauels et al 2017, Figure 5), also, for example, allowing for efforts comparing SRES and RCP scenarios, informed by both CMIP3 and CMIP5 models (Rogelj et al 2012, IPCC AR5 WGI, Fig 12.40). The reviewer refers to Figure 4 of Schwaber et al 2019 to illustrate the assumed faster temperature response under an instantaneous quadrupling of CO2 for MAGICC6. The figure, however, nicely shows that MAGICC6 is right within the range of CMIP5 models in terms of rate and

absolute level of warming, underlining the conclusion of the Schwarber et al 2019 that "the comprehensive SCMs [simple climate models] can generally replicate the long-term results of general circulation models" (Schwarber et al 2019).

AOGCM/ESM warming vs observations: The reviewer raises the concern that CMIP5 models (replicated well by MAGICC6) would overestimate observed warming and near-term temperature change, by citing IPCC WGI AR5 Figure 11.25 and Fyfe et al 2013, both published during the time of what became known as the "warming hiatus". Figure 1 shows an update of IPCC WGI AR5 Figure 11.25 (https://www.climate-lab-book.ac.uk/comparing-cmip5-observations/) created by Ed Hawkins.

Figure 1 (Ed Hawkins): Updated version of IPCC AR5 Figure 11.25b with the Had-CRUT4.6 global temperature time-series and uncertainty (black). The CMIP5 model projections are shown relative to 1986-2005 (light grey) and 2006-2012 (dark grey). The red hatching is the IPCC AR5 indicative likely range for global temperatures in the 2016-2035 period, with the black bar being the assessed 2016-2035 average. The blue lines represent other observational datasets (Cowtan & Way, NASA GISTEMP, NOAA GlobalTemp, BEST). The green axis shows temperatures relative to 1850-1900 (early-industrial period).

Numerous more recent studies have been able to show that the causes for this temporal flattening of the warming curve are well understood (Medhaug et al. 2017) and that CMIP models can be reconciled well with observations if the right methodologies are applied (Cowtan et al 2015, Tokarska et al 2019, Hausfather et al 2020). Figure 2 shows the contributions to the differences in recently observed and modelled warming results, as presented in Tokarska et al (2019).

Figure 2 (Tokarska et al 2019): Contributions to differences in recent observed and modelled warming. a,b, Time series of modelled and observed warming (a) and different effects leading to adjustments in observed and modelled GBST (b). The length of the bars (horizontal black lines in b) shows upper (lower) estimates of the influence of

Pacific variability on warming. The spread arises from uncertainty in both observations and the forced signal (effects 5 and 6), from missing years (effects 8 to 10), and reflects the range across four studies (effect 7). Vertical black lines in b indicate 5–95% uncertainty ranges. Effects indicated by an asterisk are used for the net effect (bar 4). The global mean temperature base period is 1961–1990 in a, and 2006–2015 relative to 1986–2005 in b (see Methods for details). AA, anthropogenic aerosols; S, solar; V, volcanic.

In order to increase the transparency with regard to the MAGICC6 performance compared to observations, we have revised and re-organized Figure 1 and included the most recent HadCRUT4 (HadCRUT4 - global temperature dataset) and BEST(Berkeley Earth Surface Temperature Study (BEST)) observational datasets, as well as the likely GMT response range (66% model range) for the assessed pathways:

Figure 3 (manuscript Figure 1): Historical and annual global Greenhouse Gas (GHG) emission and Global Mean Temperature (GMT) trajectories for the NDC reference scenario and two NDC improvement scenarios with a 10% and 33% reduction of 2030 emission levels relative to the NDC reference. Scenarios for Big Emitters are labeled "BE" and scenarios for all countries using "ALL". Also shown is the observed global temperature change based on HadCRUT4 (solid grey ,HadCRUT2020) and BEST (dashed grey, BEST_2020) datasets. All GMT estimates are provided relative to the 1850-1900 average. Annual global GHG emissions are provided in gigatonnes $CO_2$ equivalent (Gt CO2eq).

In this context it may be useful to note that MAGICC6 has been providing crucial science input under the UNFCCC, for example in relation to the (I)NDC assessment of the recent past (UNFCCC, 2016). The presented assessment is therefore also consistent with the scientific basis informing the current international climate negotiations under the Paris Agreement.

In summary, we hope that we have been able to demonstrate that neither MAGICC6

significantly overestimates 21st century warming rates compared to AOGCMs/ESMs, nor the AOGCMs/ESMs used for the calibration of the MAGICC model would overestimate observed warming to a degree that would lead to the presented assessment being overly pessimistic.

RC1.02: While this paper is in line with IPCC's estimate of necessary reductions from the 1.5degree special report, the report largely relies on just MAGICC6 and FAIR (another simple climate model) to make these emissions statements. FAIR is only marginally better than MAGICC6 at meeting the median CMIP5 temperature response, as shown in Schwarber et al. 2019. MAGICC6 has enough tunable parameters to produce results in-line with CMIP5 models (Hartin et al. 2015, GMD, doi:10.5194/gmd-8-939-2015; Meinhausen et al. 2011b, ACP,* doi:10.5194/acp-11-1457-2011) and, presumably, observed GMST. However, these scenarios are not explored, and represent a shortcoming in the submitted paper. As a result, the warming scenarios used in this paper likely have a noticeable hot bias.

*(Meinshausen et al. 2011b, ACP, the companion paper to the one cited; it shows comparisons of MAGICC6 being both in line with and warmer than AOGCM results under different presentations)

AR1.02: While we appreciate the continued concern of the reviewer, we would like to point to our detailed response above and also highlight that warming results of theoretical pulse response experiments (which show that MAGICC6 is actually performing rather well) are different to climate projections based on actual emission pathways. Below, we show a figure from Nauels et al (2017), that directly compares MAGICC6 global mean temperature responses to available corresponding CMIP5 projections under all RCPs extended up to 2300, the timescale of concern for our analysis.

Figure 4 (Nauels et al 2017): Global mean temperature (GMT) projections until 2300 for all RCP extensions based on the historically constrained probabilistic MAGICC setup; 90% ensemble range in light colors, 66% ensemble range in darker colors, medians as

solid lines. Available CMIP5 GMT reference time series are shown as thin black lines. All temperature projections are given relative to 1850.

Indeed, MAGICC6 has enough tunable parameters and it is able to capture the CMIP5 response accurately. Because this capability has been explored in previous work, there is no reason to believe that MAGICC6 has a significant hot bias (see Figure 3). Hence, this study focuses on pathways that allow to assess the longer-term implications of reducing 2030 emission levels.

In order to be more clear about the capability of MAGICC6 to capture the responses from more complex models, we have added the following sentence in line 115: "Previous studies have shown that MAGICC6 captures more complex model responses well (Meinshausen et al 2011, Nauels et al 2017), while overarching efforts to reconcile observations with complex model responses have been successful as well (Cowtan et al 2015, Tokarska et al 2019, Hausfather et al 2020)."

RC1.03: The consideration above would, of course, bring into question the core proclamation that none of the suggested NDC improvement scenarios meet the 1.5 degree goal. If the actual climate system does not warm as much under the ALL33 scenario (or any other scenario) as suggested by the MAGICC6 model runs, the likelihood of meeting the 1.5 degree goal would be much higher than suggested in the submitted paper. This criticism is not to suggest that the core message will change dramatically if the potential hot bias is accounted for – it could still very well be that the ALL33 scenario does not end below the 1.5 degree goal, or does end below 1.5 degrees but not strongly enough to say it would do so with sufficiently high confidence. However, for this paper to be considered for publication, more justification (and/or context) needs to be given for the amount of warming that results from each emissions scenario, with careful attention to how well MAGICC6 can simulate observed warming.

AR1.03: We thank the reviewer for elaborating on this further. By showing that MAGICC6 provides robust and consistent projections when comparing its output to CMIP

results, we hope that the reviewer's concern that this study may provide overly warm projections could be addressed.

In addition, we have adjusted Figure 1 (please see AR1.01) to address the referee's concern and increase transparency in terms of the uncertainties related to MAGICC. We have now included likely ranges in Figure 1, with the intention to highlight the probabilistic nature of the MAGICC projections as well as the wide range of uncertainties underlying these estimates. Like this, we hope to also visualise more clearly that projected warming could still be significantly lower, or significantly higher than the median "best estimate".

As already mentioned in RC1.01, we have now also included the HadCRUT4 observational dataset in Figure 1 to allow for a direct comparison of MAGICC6 output and observations, noting the issues with the blended-masking temperature metric applied in HadCRUT4 that is discussed in great detail e.g. in Tokarska et al (2019).

RC1.04: On one hand, if this particular configuration of MAGICC6 does definitively have a hot bias (as indicated by papers cited above), its use could be justified by stating that application of a hot model means any emissions scenario that meets the 1.5 degree goal under this analysis should meet that temperature goal no matter what model (or ensemble of models) is considered. That is, intentional underestimation in an emission scenario's potential success has the policy benefit of higher confidence in that scenario overall. On the other hand, statements such as the penultimate sentence of the abstract (lines 15-17), the opening paragraph of section three (lines 156-159), or the "clear evidence" conclusion (line 257) should probably be made based on likely warming scenarios, not on warmer-than-average scenarios.

AR1.04: We thank the reviewer for providing a rationale that incorporates the rationale of probabilities for achieving certain climate targets. As stated above, the existence of a large array of different uncertainty sources means that there is a chance of overestimating (or underestimating) future warming with the presented approach, as well as

all existing other modelling approaches. However, we are confident that we are pro-
viding an up-to-date and robust set of temperature projections. We would not consider
publishing results knowing that the underlying modelling would be biased and likely
overestimate warming, just in order to be on the "safe side" when it comes to the policy
context. We have revisited the statements highlighted by the reviewer in order to not
provide the impression that we are overly confident in our results. Lines 15-17 now
read: "An increase in aggregated NDC ambition of big emitters by 33% in 2030 does
not reduce presented climate impacts by more than about half compared to limiting
warming to 1.5°C." Lines 156-159 now read: "Applying a consistent model setup, our
slightly more optimistic "current NDC" reference scenarios used here would lead to a
warming reduction by about 0.2°C, implied by a reduction in 2030 GHG emission levels
of about 1.5 Gt CO2eq compared to the 2018 CAT pathway." Line 257 now reads: "Our
results show the need for a transformational increase in 2030 ambition by countries to
achieve the 1.5°C limit and to avoid the impacts of exceeding this level of warming."

Minor Comments

RC1.05: Line 112 – spacing

AR1.05: The spacing in line 112 has been fixed.

RC1.06: Lines 136 to 140 – sentences are a little hard to follow

AR1.06: Lines 136 to 140 have been restructured to allow for better readability. "We
present changes in hot extremes as land fraction distributions of changes in the inten-
sity of the hottest day in a year (TXx), following the method introduced by Fischer et
al (2013). This analysis is based on a time-slicing approach: for each model, a 21-
year time period is selected for which the averaged global warming corresponds to the
end-of-century GMT increase given by the respective scenarios (see Table 1). At each
grid-cell, the hottest days of a year (TXx) are averaged over the selected 21-year pe-
riod and these averages of all selected CMIP5 runs are merged into one TXx change
distribution per scenario and region."

RC1.07: Line 165 – verb confusion

AR1.07: The sentence was simplified. It now reads: "Scenarios in which emission reductions are limited to big emitters only lead to higher warming levels."

RC1.08: Lines 173 to 176 – I don't follow this sentence at all

AR1.08: The corresponding paragraph was revised. Line 173 to 177 now reads: "In addition to the GMT trajectories over the 21st century, assessments of related climate impacts are provided. These additional impacts are provided for three scenarios: the "current NDC" ambition reference pathway, the 33% reduction of 2030 emissions from the "current NDC" ambition pathway for big emitters (BE33) and a representative 1.5°C pathway (based on SSP1-RCP1.9). This allows for the comparison of impacts implied by the three scenarios ranging from the current ambition level to a Paris compatible level of ambition (1.5°C)."

RC1.09: Line 180 – the unit "cm" is separate from its numbers on the previous line

AR1.09: Thanks, this has been fixed.

RC1.10: Line 181 to 182 – the two parts of the sentence are ordered backwards from what a reader might normally expect

AR1.10: The two parts of the sentence have been swapped, the sentence was reformulated.

RC1.11: Table 2, NDC column, 2300 row – spacing

AR1.11: Thanks, Table 2 has been reformatted!

RC1.12: Lines 219 to 220 – the last phrase here feels awkwardly worded

AR1.12: The whole paragraph has been restructured. Lines 218 to 222 now read: "As expected, lower emission targets for big emitters have the biggest effect on GMT and related impacts. Therefore, scenarios for big emitters only show relatively similar

temperature responses for high GMT trajectories compared to scenarios for all countries. However, the increasing differences in temperature outcomes between these two scenario groups show that, with increasing ambition, the relative importance of the contributions from small emitters grows (compare Table 1)."

RC1.13: Line 245 – unnecessary comma after first word; also "...each country's specific..."(possessive)

AR1.13: Both errors have been corrected.

RC1.14: Line 249 – unnecessary capitalization ("Land use")

AR1.14: Thanks, has been corrected.

RC1.15: Overall – aside from the major reservation about warming bias in the chosen model this is a very well-written and well-reasoned paper.

AR1.15: Thank you very much for this positive feedback. We have tried to address the warming bias concerns comprehensively and hope that, in combination with the presented extensive revisions, the manuscript has been improved to the reviewer's satisfaction.

References: Berkley: Berkeley Earth Surface Temperature Study (BEST), http://berkeleyearth.lbl.gov/auto/Global/Land_and_Ocean_complete.txt, 2020.

Climatic Research Unit (University of East Anglia): HadCRUT4 - global temperature dataset, https://crudata.uea.ac.uk/cru/data/temperature/, 2020.

Cowtan, K., Hausfather, Z., Hawkins, E., Jacobs, P., Mann, M. E., Miller, S. K., Steinman, B. A., Stolpe, M. B., and Way, R. G.: Robust comparison of climate models with observations using blended land air and ocean sea surface temperatures, Geophysical Research Letters, 42, 6526–6534, https://doi.org/10.1002/2015GL064888, 2015.

Hausfather, Z., Drake, H. F., Abbott, T., and Schmidt, G. A.: Evaluating the Performance of Past Climate Model Projections, Geophysical Research Letters, 47, e2019GL085 378,https://doi.org/10.1029/2019GL085378, https://agupubs.onlinelibrary.wiley.com/doi/abs/10.1029/2019GL085378, e2019GL085378 2019GL085378, 2020.

Medhaug, I., Stolpe, M. B., Fischer, E. M., and Knutti, R.: Reconciling controversies about the 'global warming hia- tus', Nature, 545, 41–47, https://doi.org/10.1038/nature22315, http://www.nature.com/doifinder/10.1038/nature22315, 2017.

Meinshausen, M., Raper, S. C. B., and Wigley, T. M. L.: Emulating coupled atmosphere-ocean and carbon cycle models with a simpler model, MAGICC6 - Part 1: Model description and calibration, Atmospheric Chemistry and Physics, 11, 1417–1456, https://doi.org/10.5194/acp-11-1417-2011, ttp://www.atmos-chem-phys.net/11/1417/2011/, 2011.

Nauels, A., Meinshausen, M., Mengel, M., Lorbacher, K., and Wigley, T. M. L.: Synthe- sizing long-term sea level rise projections – the MAGICC sea level model v2.0, Geo- scientific Model Development, 10, 2495–2524, https://doi.org/10.5194/gmd-10-2495- 2017, https://www.geosci-model-dev.net/10/2495/2017/, 2017.

Rogelj, J., Meinshausen, M., and Knutti, R.: Global warming under old and new scenarios using IPCC climate sensitivity range estimates, 2, https://doi.org/10.1038/NCLIMATE1385, 2012.

Schwarber, A. K., Smith, S. J., Hartin, C. A., Vega-Westhoff, B. A., and Sriver, R.: Evaluating climate emulation: fundamental impulse testing of simple climate mod- els, Earth System Dynamics, 10, 729–739, https://doi.org/10.5194/esd-10-729-2019, https://www.earth-syst-dynam. net/10/729/2019/, 2019. Tokarska, K. B., Schleussner, C.-F., Rogelj, J., Stolpe, M. B., Matthews, H. D., Pfleiderer, P., and Gillett, N. P.: Rec- ommended temperature metrics for carbon budget estimates, model evaluation and climate policy, Nature Geoscience, 12, 964–971, https://doi.org/10.1038/s41561-019- 0493-5, https://doi.org/10.1038/s41561-019-0493-5, 2019.

UNFCCC: Aggregate effect of the intended nationally determined contributions : an update, Tech. Rep. FCCC/CP/2016/2, UNFCCC, 2016.

[Figure]

[Figure]

**Fig. 1.** Figure 1 (Ed Hawkins): Updated version of IPCC AR5 Figure 11.25b with the Had-CRUT4.6 global temperature time-series and uncertainty (black). The CMIP5 model projections are shown relative to 1986-2005

[Figure]

**Fig. 2.** Figure 2 (Tokarska et al 2019): Contributions to differences in recent observed and modelled warming. a,b, Time series of modelled and observed warming (a) and different effects leading to adjustments

**Fig. 3.** Figure 3 (manuscript Figure 1): Historical and annual global Greenhouse Gas (GHG) emission and Global Mean Temperature (GMT) trajectories for the NDC reference scenario and two NDC improvement scenari

[Figure]

**Fig. 4.** Figure 4 (Nauels et al 2017): Global mean temperature (GMT) projections until 2300 for all RCP extensions based on the historically constrained probabilistic MAGICC setup; 90% ensemble range in light

---

## Author Comment (AC2) · 9 Mar 2020

Geiges et al.: "Incremental improvements of 2030 targets insufficient to achieve the Paris Agreement goals" https://doi.org/10.5194/esd-2019-54 Compiled author responses to reviewer 2 comments

REVIEWER #2

General Comment

Reviewer Comment (RC2.00): In this paper, the authors first estimate emissions related to Nationally Determined Contributions scenarios, and then calculate correspond-

ing climate impacts. The paper is well written albeit brief in the analysis. The study design is particularly interesting and novel and the methods used are comprehensive. Presentation of the climate impacts and results in general, could be improved. The results section is shorter than the methods. Discussion could be made more interesting. Overall, this will be a very good paper, but I think there are some changes that are within the authors reach to greatly improve the work. Criticisms below.

Author Response (AR2.00): We would like to thank the reviewer very much for these thorough and constructive comments. We believe that by addressing them, we were able to improve the submission a lot. We have revised all figures, and expanded the results and discussion section, following the referee's suggestion. We provide more detailed responses to the individual comments below.

Major Comments

RC2.01: The authors appear to have made no efforts to make neither the results nor code available. I urge the authors to make at least the additional data used to make the figures available. Additionally, data on the NDC emissions scenarios and corresponding term-perature outcomes from MAGICC would also be useful to the community (e.g. Fig1).

AR2.01: Thank you for raising this important issue! We have made the results and the necessary plotting routines available in an online repository. A data and code availability section has been added to the manuscript, reading: "All data and code underlying the results presented in the figures of the study can be found here: https://gitlab.com/ageiges/ndc_ambition_esd_paper_material"

RC2.02: One weakness, given the substantial work already done, is why not more climate impacts considered. The authors have gone to great efforts in the first half of the work relating to NDCs, then only present three first order climate impacts. In the paper the authors estimate sea level rise impacts, extreme temperatures and economic damages – from different models/approaches. Considering precipitation, both high

and low indicators, would have surely been straightforward to add to the statistics on extreme temperatures, for example.

AR2.02: We thank the referee for these suggestions. However, we would like to point out that the scope of our analysis focussing on the different updated NDC pathways. The three selected impacts have been chosen to illustrate the consequences of those pathways for different systems: extreme events, long-term changes and human systems. We agree that this could be motivated better in our paper and added the following paragraphs to section 2.4 of the manuscript (starting from from line 125): "Global mean temperature is an established metric that allows to approximate a range of different climate impacts. However, regional or sectoral changes can be much more pronounced than those in global mean temperature. As established in the IPCC SR1.5, a wide range of vulnerable systems are sensitive to temperature differences of 0.5°C or less, including terrestrial and marine ecosystems such as coral reefs, cryosphere changes and extreme weather indices \citep{Schleussner2016b}. Providing a comprehensive analysis of the differences in climate impacts would go beyond the scope of this analysis that focuses on the implications of different 2030 NDC update scenarios. However, to illustrate the relevance of these differences for a range of different climate impacts, we have chosen three exemplary impact indicators: an extreme event indicator (extreme hot days), a time-lagged response (long-term sea level rise), and economic damages."

With regard to the specific suggestion on adding precipitation, we note that precipitation changes are less robust than other variables, at least on the global scale. Extreme precipitation is scaling more or less linear with GMT increase, but the drought signal is very regionally dependent and linked to circulation changes (compare e.g. Schleussner et al. 2016). Therefore, we are of the view that a precipitation analysis would require substantial additional effort, including a more detailed analysis on the regional level which we consider out of scope of the paper. We, however, very much agree that it is exactly that differentiation in terms of differential impacts per warming level that is

required to inform the policy discourse around NDC updating and hope to address this important question in future research.

RC2.03: Figure 4 is nicely designed, essentially useless. Even if one could accurately read such small graphs, knowing the change in probability density function conveys very little information. You have made great efforts to make the first half of the paper policy-relevant – linking to NDCs etc – and then the way the Temperature and SLR impacts are presented and discussed is without value. At least CDFs could be used to show% of land impacted, or population impacted. In any case the figures are so small they cannot be used – the only point is to demonstrate that you have the information but have no intention for other people to use it.

AR2.03: We thank the reviewer for the comment on this figure and for the suggested improvements. We revised the figure (now Figure 3) to show CDFs (as suggested by the reviewer), increased the size of the global panel (lower left corner), and added some interpretation guidance. For instance, we better capture the land area affected by an increase in TXx by at least 2°C with dashed lines going from the 2°C change (vertical dashed gray line) towards the left y-axis. In the global panel, these land area fractions are labelled next to the y-axis, in all other regional panels, only the corresponding lines are shown. These additions should help the reader to understand how the CDF graphs have to be interpreted and hopefully contribute to the usefulness of the figure.

We agree that the regional panels are relatively small. It is however technically difficult to increase their size. For many stakeholders, impact projections only become relevant on a regional level (not globally). Therefore, we want to keep the regional panels in the figure. While the global panel is large enough to be read and interpreted in detail, the regional panels can be compared with the global panel to get an idea about regional differences in heat extreme projections.

Figure 1 (manuscript Figure 3): Changes in hot extremes (TXx) in the period 2081-2100 relative to 1986-2006 for the NDC scenario (red), the BE33 scenario (yellow) and the

1.5°C scenario (cyan). Changes are presented as land area fraction (y-axis) affected by changes in TXx (x-axis) for 26 world regions and globally. Cumulative distributions are based on an area weighted aggregation of all TXx change values projected at grid-cells within a region and across climate models. A vertical dashed gray line indicates TXx changes of 2°C. The asterixis and the horizontal lines going from the 2°C line to the left y-axis show the land fraction that is affected by an increase in TXx of at least 2°C.

RC2.04: Perhaps an illustrative diagram in section 2.1 would help describe the methods. I'm not convinced by using Txx – hottest day of the year, because I am unconvinced that theGCMs are able to consistently, especially across regions, reproduce the single hottest day each year. Why pick such a difficult target, when perhaps the 5th hottest day each year, or a p99 over the 21-year period would probably equally be sufficient to estimate the changing temperature distribution and is likely much better reproduced by the models?If you insist on this indicator, can you provide some validation that at least the GCMsused accurately reproduce Txx for the historical period with error less than 0.6 degC –because that's the difference between your BE33 and 1.5C scenarios.

AR2.04: We chose TXx as an absolute measure of heat extremes as it is a metric with a very simple definition that is very common in the climate impact literature. Furthermore, GCMs have been shown to reproduce TXx changes well (compare e.g. Sillmann et al. 2013, Seneviratne et al. 2016, Schleussner et al. 2017 and the IPCC AR5 WG1 and SR1.5).

For instance, the section about hot extremes in the IPCC 1.5C Special Report also chose to present changes in TXx in the same regions we are using here (please see Figure 2). Our revised figure (now Figure 3) clearly shows how differences of 0.5°C (between 1.5°C and 2°C) can be distinguished on the regional level. In other work, we have shown how GMT differences down to 0.2°C can be meaningfully differentiated in this indicator (Pfleiderer et al. 2018). As a validation against observations, we

also include Figure 3 below, comparing a 0.5°C difference in observations and CMIP5 model output, clearly showing significant differences.

Please note as well that the CDF's (previously PDF's) shown in the new Figure 3 are a highly aggregated result of all used CMIP5 models and averaged over 20 years. We are quite confident that a twenty year average of yearly temperature extremes is a robust metric and that the signal of climate change as illustrated in Figure 4 can be clearly distinguished from internal variability.

Figure 2: the IPCC 1.5C Special Report also chose to present changes in TXx

Figure3: Comparison of a 0.5°C difference in observations and CMIP5 model output

RC2.05: Figure 3 – why is 1.5 in red, and NDC in Blue? This doesn't make sense and is opposite to Fig 4 and Fig 2 colour schemes. Are there no uncertainty ranges associated with the economic impacts assessment from the Burke methodology?

AR2.05: We thank the reviewer for pointing out this oversight in Figure 3 (now Figure 4). The color scheme has been updated and is now consistent with the other figures.

Burke et al. (2018) report four sources of uncertainty that arise in their analysis, namely from SSPs, discount rates, regression (bootstrapping) and the climate models. Our analysis uses SSP 2 and does not test for different discount rates, therefore leaving two sources of uncertainty.

Our new Figure 4 now shows boxplots that capture uncertainty from the GCMs (using median bootstrap estimates for each GCM). We thank the reviewer for this very valuable suggestion that contributes to a more robust representation of the economic impacts:

Figure 4 (manuscript Figure 3): Economic damages under different scenarios of GMT increase. The boxplots contain estimates for different GCMs, and show the percentage difference between GDP per capita under selected temperature pathways (no change in current NDCs, BE33 scenario resembling a 33\% change in NDCs of big

emitters and a 1.5°C pathway) and GDP per capita under a no climate change scenario. The lower and upper hinges correspond to the first and third quartiles (the 25th and 75th percentiles). The upper whisker extends from the hinge to the highest value that is within 1.5 × the interquartile range of the hinge. Estimates are given for mid-century (2046-2065) and end-of-century (2081-2100). Countries are grouped by either geographical regions (South Asia and Latin America) or political groupings following the UN classifications (Small Island Development States, SIDS and Least Developed Countries, LDCs).

RC2.06: It's not clear what is the point of the including these climate impacts. The results are presented in a mechanical fashion. Why they are included, and justification for the specific impacts, is not provided. In the discussion there are only 4 lines about them. Try to explain the "so what" for the reader. What does 1m sea level rise mean? Does 1-3degC really make a difference for the hottest day of the year. It doesn't sound like much to regular people, but if you're an expert, we know it has impacts on animals, vulnerable people, chance of crop failure, labour productivity, power plant efficiencies, peak electricity demands, rail tracks and roads – and on and on and on. So try and bring this perspective to the reader, on why 1-3 degC change for the hottest day of the year is significant. The same applies to sea level rise that is in the order of centimeters... doesn't sound like much, but if you live in the Netherlands, or Bangladesh, or Miami – it's terrible news!

AR2.06: We thank the referee for this suggestion. We have substantially expanded the section motivating our impact selection (see AR2.02) as well as the discussion of the implications of these impact differences.

We added the following text to the discussion section from line: "The IPCC Special Report on 1.5°C has identified a range of key reasons for concern as high above 1.5°C including by extreme weather events, for unique and threatened systems as well as globally unequal impacts (Hoegh-Guldberg et al 2018). Our results confirm these findings and add additional information also in comparison with trajectories implied by

current NDCs. Differences in median sea level rise between a 1.5°C and NDC scenario in 2100 amount to about 20cm, as much as the world has experienced over the observational period, which has already contributed significantly to the occurrence of coastal flooding (IPCC 2019). Beyond 2100, the difference could amount to almost one meter or more until 2300 (compare Table 2). For extreme heat, we find a doubling in impacts between 1.5°C and NDC pathways (compare Figure 4), with profound regional differences. For Central Europe, for example, we project that about 40% of the land-area would experience a TXx increase under current NDC pathways of about 4°C. For tropical regions, exceeding a new climate regime in terms of extreme temperatures will be reached already for 2°C warming. Exceeding 1.5°C will also substantially increase the risks of exceeding tipping points of the earth system going forward (Schellnhuber and Rahmstorf 2016)."

RC2.07: Supplementary information is quite concise and no results provided in data form.

AR2.07: We agree, but do not see the need to substantially extend the currently provided Supplement. All data shown in the figures is now also available on the new gitlab repository, included in the data and code availability section: https://gitlab.com/ageiges/ndc_ambition_esd_paper_material

Minor Comments

RC2.08: Line 79: Spell out AR5

AR2.08: Line 79 has been changed accordingly.

RC2.09: Line 198-199 – you say 4degC here – but you should clarify that this is for the hottest day only, not mean temperature rise, as it could easily be misunderstood.

AR2.09: We have changed the corresponding paragraph in the manuscript to read: "Under the NDC scenario, changes in TXx would be most pronounced. The global median increase of TXx under this scenario is projected to be 2.7°C above the 1986-

2005 level. The increase in the high-end tail is most pronounced under this scenario with 10\% of the land area experiencing increases in TXx of over 4°C above the 1986-2005 levels."

RC2.10: Line 231: You say results in line with other studies, yet you only provide one citation? Consider works that could back up this statement e.g. by Piontek et al (2014, PNAS), Scheussner et al (2016, ESD), Byers et al (2018, ERL), Mora et al (2018, Nat CC)

AR2.10: We thank the referee for pointing that out and have expanded the reference list.

References: Knutti, R., Sedláček, J. Robustness and uncertainties in the new CMIP5 climate model projections. Nature Clim Change 3, 369–373 (2013). https://doi.org/10.1038/nclimate1716

Burke, M., Davis, W. M., and Diffenbaugh, N. S.: Large potential reduction in economic damages under UN mitigation targets, Nature, 557, 549–553, https://doi.org/10.1038/s41586-018-0071-9, https://doi.org/10.1038/s41586-018-0071-9, 2018.

Ove Hoegh-Guldberg, Jacob, D., Taylor, M., and al, E.: Impacts of 1.5oC global warming on natural and human systems, in: Global Warming of 1.5 C :An IPCC special report on the impacts of global warming of 1.5 C above pre-industrial levels and related global greenhouse gas emission pathways, in the context of strengthening the global response to the threat of climate change" https://doi.org/10.1093/aje/kwp410, 2018. Schellnhuber, H. J. Rahmstorf, S. and Winkelmann, R.: Why the right climate target was agreed in Paris, Nature Clim. Change, 6, 649–653, https://doi.org/10.1038/nclimate3013, http://dx.doi. org/10.1038/nclimate3013, 2016.

Schleussner, C.-F., Lissner, T. K., Fischer, E. M., Wohland, J., Perrette, M., Golly, A., Rogelj, J., Childers, K., Schewe, J., Frieler, K., Mengel, M., Hare, W., and Schaeffer,

M.: Differential climate impacts for policy-relevant limits to global warming: the case of 1.5°C and 2°C , Earth System Dynamics, 7, 327–351, https://doi.org/10.5194/esd-7-327-2016, https://www.earth-syst-dynam.net/7/327/2016/, 2016.

Schleussner, C.-F., Pfleiderer, P., and Fischer, E. M.: In the observational record half a degree matters, Nature Climate Change, 7, 460–462, https://doi.org/10.1038/nclimate3320, http://dx.doi.org/10.1038/nclimate3320http://www.nature.com/doifinder/10.1038/nclimate3320, 2017.

Sillmann, J., Kharin, V. V., Zhang, X., Zwiers, F. W., Bronaugh, D., Zhang, X., Bronaugh, D., Zwiers, F. W., and Bronaugh, D.: Climate extremes indices in the CMIP5 multimodel ensemble: Part 2. Future climate projections, Journal of Geophysical Research: Atmospheres, 118, 2473–2493, https://doi.org/10.1002/jgrd.50188, http://doi.wiley.com/10.1002/jgrd.50188http://onlinelibrary. wiley.com/doi/10.1002/jgrd.50188/abstract, 2013.

[Figure]

[Figure]

**Fig. 1.** Figure 1 (manuscript Figure 3): Changes in hot extremes (TXx) in the period 2081-2100 relative to 1986-2006 for the NDC scenario (red), the BE33 scenario (yellow) and the 1.5°C scenario (cyan). Change

[Figure]

**Figure 3.5 |** Projected changes in annual maximum daytime temperature (TXx) as a function of global warming for IPCC Special Report on Managing the Risk of Extreme Events and Disasters to Advance Climate Change Adaptation (SREX) regions (see Figure 3.2), based on an empirical scaling relationship applied to Coupled Model Intercomparison Project Phase 5 (CMIP5) data (adapted from Seneviratne et al., 2016 and Wartenburger et al., 2017) together with projected changes from the Half a degree additional warming, prognosis and projected impacts (HAPPI) multimodel experiment (Mitchell et al., 2017; based on analyses in Seneviratne et al., 2018c) (bar plots on regional analyses and central plot, respectively). For analyses for other regions from Figure 3.2 (with asterisks), see Supplementary Material 3.SM.2. (The stippling indicates significance of the differences in changes between 1.5°C and 2°C of global warming based on all model simulations, using a two-sided paired Wilcoxon test (P = 0.01, after controlling the false discovery rate according to Benjamini and Hochberg, 1995). See Supplementary Material 3.SM.2 for details.

**Fig. 2.** Figure 2: the IPCC 1.5C Special Report also chose to present changes in TXx

[Figure]

Figure 2 | Historical 0.5 °C warming is representative for 1.5 °C versus 2 °C differences. Changes in hot extremes (**a**) and extreme precipitation (**b**) due to 0.5 °C warming over the historical period (purple) and between 1.5 °C and 2 °C (grey) as simulated in an ensemble of CMIP5 models. Model-specific time slices are derived to match historical 0.5 °C warming up to the 1991–2010 reference period and future warming levels of 1.5 °C and 2 °C above pre-industrial conditions (see Supplementary Information). The filled envelope depicts the 5–95% ensemble range and thin lines represent individual models. The observed differences are given for comparison in blue and red as in Fig. 1.

**Fig. 3.** Figure3: Comparison of a 0.5°C difference in observations and CMIP5 model output

**Impact of climate change on GDP**

Latin America

LDCs

SIDS

South Asia

% change in GDP per capita

Year

Temperature pathway ~1.5°C BE33 NDC

**Fig. 4.** Figure 4 (manuscript Figure 3): Economic damages under different scenarios of GMT increase. The boxplots contain estimates for different GCMs, and show the percentage difference between GDP per capita

---

## Author Response (AR2)

*Geiges et al.: "Incremental improvements of 2030 targets insufficient to achieve the Paris Agreement goals"*
https://doi.org/10.5194/esd-2019-54
*Compiled author responses to reviewer comments*

**REVIEWER #1**

*(1) General comments/suggestions*

**RC1.00:** This revised paper begins by translating the Nationally Determined Contributions from the Paris Agreement into emissions scenarios, also creating several mitigation emissions scenarios based on the NDC emissions scenarios. Then, using the simple climate model MAGICC6, the paper creates projections of global surface temperature and examines a sampling of consequences to that warming climate. The revision is in response to two initial referee reviews, both of which appreciating the study's design, the results presented, and the paper's readability, while both also having requests for major revisions.

Referee 1 expressed concern over the use of MAGICC6 as representative warming in the future, and referee 2 offered varied criticism on the choices and depth of climate consequences examined and how the data was presented. The authors resisted many of these concerns, although a large amount of new prose and a number of figure alterations now appear in response most (but not all) of referee 2's criticisms in the revised paper.

Some further discussion concerning the authors' responses to referee 1 are presented below. This issue between the authors and referee 1 may not be able to be resolved. Even if that is the case, some notable revisions would still be suggested before this paper would be ready for acceptance. While one sentence was added to section 2.3 of the paper in response to referee 1, there is enough material in AR1.01 and AR1.02 that an entire paragraph could and should be added to section 2.3 instead so the authors can solidly assure future readers of their confidence in MAGICC6. In a similar fashion, a sentence or two based on material from AR2.04 should be added to section 2.4.2 to further justify that this quantification of extreme warmth (TXx) is a robust measurement.

**AR1.00:** *The authors would like to thank the reviewer very much for this second constructive review round and are pleased that the presented work is generally*

*considered to have a very good ESD suitability. We regret to not have added a more in-depth discussion of the climate model choice and caveats in the first response round and are happy to introduce additional information on the applicability of the simple climate carbon-cycle model MAGICC version 6 in the manuscript, as requested by the referee. We hope that the revisions now fully address the raised concerns.*

*(2) Detailed responses*

**RC1.01:** There seems to be differing interpretations of Figure 4 of Schwarber et al. 2019. While the authors do note that MAGICC6 (orange line) falls within the range of CMIP5 results (grey lines), the figure caption points out that it is the bold grey lines that show CMIP5 models whose ECS values (range 2.5-3.5°C) are similar to the ECS (3°C) driving the MAGICC6 run. Of these bold grey lines, all but one fall below the orange line of MAGICC6. While Schwarber et al. do state, as the authors quote, that "the comprehensive SCMs can generally replicate the long-term results of general circulation models", and the SCM results do fall within the overall range GCM results, that "general" replication fails somewhat when more specifics are considered and the SCMs still fall within the upper half of GCM results.

The focus on the hiatus period in the second half of AR1.01 is less important – and less certain – than the authors argue. In particular, the provided figure from Tokarska et al. shows very high uncertainty in the estimate of cooling from Pacific variability and moderate uncertainty in the effect of changed forcings, with difficulty in comparing the forcing values' effects due to the different combinations of stratospheric volcanic aerosols, solar variation, and anthropogenic aerosols. All options in that category consider volcanic stratospheric aerosol optical depth as part of their changed forcing, but there is some doubt on the actual strength of SAOD cooling. Chylek et al. 2020 (doi: 10.1029/2020GL087047) provides some evidence that SAOD cooling is lower than previously suggested, which would reduce the effect minor volcanoes have had on the hiatus unless the climatic SAOD response is highly nonlinear. Even if the hiatus was understood with the confidence suggested by the authors, particularly with the forcing corrections, it is worth noting that Tokarska figure also shows how the newer generation of CMIP6 models does an even worse job of matching observed warming than the CMIP5 models did, with a slight positive difference between CMIP6 and CMIP5 over the 2006-2015 time period relative to 1986-2005, a fact closely mirrored in the Discussion section of Tokarska et al. 2020 (doi: 10.1126/sciadv.aaz9549).

**AR1.01:** *We would like to thank the referee for the detailed feedback on the information included in the last review round on the applicability of MAGICC6. We have tried to*

*further accommodate the referee's concerns and extensively revised section 2.3. It now reads:*

*The constructed GHG emission pathways (following AR4 global warming potentials) are then used to derive probabilistic projections for the global mean temperature (GMT) with the reduced complexity carbon cycle and climate model MAGICC6 \citep{Meinshausen2011}.*

*MAGICC is an emulator which is calibrated against data from complex General Circulation Models (GCms), including climate sensitivity and carbon cycle information. Several studies have shown that model version 6 is able to capture both CMIP3 models\citep{Meinshausen2011} and CMIP5 models \citep{Rogelj2014, Nauels2017a} responses well, also reflecting the climate sensitivity range assessed by IPCC AR5 \citep{Rogelj2014} and the C4MIP carbon cycle response range \citep{Friedlingstein2014}.*

*The MAGICC emulations reflect the complex model response ranges for the assessed scenarios in the calibration datasets, in particular the Representative Concentration Pathways (RCPs). \cite{Schwarber2019} showed that outside this calibration space, e.g., for the response under an instantaneous quadrupling of CO2, simplified models like MAGICC cannot reproduce the same level of complexity in their response as comprehensive climate models. However, since the presented analysis of NDC pathways is well within the range of the scenarios used for the calibration, we consider MAGICC6 a suitable tool for our analysis*

*In this context it is worth noting that overarching efforts to reconcile observations with complex model responses have been successful as well \citep{Cowtan2015,Hausfather2020}. Numerous studies have been able to show that CMIP models can be reconciled well with observations if the right methodologies are applied \citep{Cowtan2015, Tokarska2019, Hausfather2020}.*

*To allow for the comparison with a representative 1.5˚C consistent pathway, MAGICC6 is also forced with SSP1-RCP1.9 type emissions \citep{Rogelj2018a}, normalized to the year 2010 emissions of the CAT pathways to ensure a consistent experimental setup. The MAGICC model is run from 1750 to 2300 to also shed light on longer-term impacts like global mean sea-level rise. It is important to stress that for this longer-term outlook, radiative forcing levels for the 22\textsuperscript{nd} and 23\textsuperscript{rd}centuries are held constant at 2100 levels for all pathways, making the post-2100 model responses more stylized than for the 21\textsuperscript{st} century.*

**RC1.03:** On one final side note, both referee 1's figure 11.25 from AR5 and the authors' figure from Tokarska et al. 2019 have the drawback that the baselines used in each are relatively recent. Since comparisons to the Paris Agreement goals of 2.0 and 1.5°C are based on the preindustrial baseline, these figures both make the model projections

appear closer to observed temperature than they would if all temperature data was instead presented on the preindustrial baseline.

*AR1.03: The 1986-2005 reference period is in line with the approach taken in the IPCC AR5 that has informed the Paris Agreement. As outlined in Tokarska et al. 2019, it is advisable for analysis in the context of the warming levels of the Paris Agreement to deploy a Paris Agreement compatible metric - which is what we have done here. It is also worth highlighting that the MAGICC model has been shown to reproduce the observational record well.*

**(3) Minor issues & corrections**

**RC1.04:** Line 24: extraneous comma
*AR1.04: Has been removed, thanks!*

**RC1.05:** Line 37: extraneous comma or "and" (hard to tell which)
*AR1.05: The 'and' has been removed.*

**RC1.06:** Line 62-64: slightly awkward compounding of phrases; might need either some readjustment or need to be split into two sentences
*AR1.06: Thanks! The phrase has been split in two parts.*

**RC1.07:** Line 68: Footnote 2 is awkwardly worded. Also, the acronym LULUCF appears in the footnote, but the acronym itself is not spelled out until line 75. This likely will not be an issue for most readers, but the ordering seems a bit odd.
*AR1.07: The footnote has been rephrased and LULUCF has been spelled out.*

*RC1.08:* Line 148: First appearance of the acronym GMT, which is not defined. I assume this means "Global Mean Temperature" and likely should be first defined in section 2.3
*AR1.08: Thanks for spotting this, GMT has been introduced in section 2.3.*

*RC1.09:* Line 187: "1.5C" is missing the degree symbol
*AR1.09: The degree symbol has been added, thanks!*

*RC1.10:* Line 191: same as above
*AR1.10: Done!*

**RC1.11:** Figure 3: The addition of the horizontal guidelines on the new CDFs are relatively clear, much more so than the PDFs in the previous version, but the figure is still hard to read. It was previously mentioned that increasing the size of the regional panels would be "technically difficult". If the difficulty is fitting the regional panels above their respective regions, it might be easier to move the panels off and around the map itself and use line segments to connect each panel to its region; the map could also be scrapped altogether if the panel labels were the full name of each region instead of abbreviations, especially as those abbreviations are not spelled out elsewhere in the paper.

*AR1.11: We thank the reviewer for commenting on this figure again and we agree that it is hard to read details from the CDFs in the regional panels. Before replying to the technical aspects of the presentation of this figure, we want to highlight that the purpose of displaying (relatively small) regional panels on a global map is to allow the reader to get an overview over regional changes in TXx. In our opinion this overview is best realised by panels that are displayed at the matching location on a world map.*

*As we agree that the readability of the regional panels is somehow limited, we tried out different ways of arranging the regional panels. The suggestion of moving the panels around a smaller world map is what we actually started with before realizing that placing the panels on the map is more compact and allows for larger regional panels. Removing the map altogether or displaying the results as a table wouldn't allow the reader to get an overview over regional differences by just looking at the figure. Please note that if the reader really wants to know all the details about the regional results he can look them up in our provided supplementary material (link). We added a direct reference in the paragraph to guide the interested reader.*

*In addition, we opted for a similar presentation of results in previous studies and got mostly positive feedback for the combination of regional panels on world maps and detailed information in the supplementary information (see* [https://www.nature.com/articles/s41558-019-0555-0](https://www.nature.com/articles/s41558-019-0555-0)*). Hopefully, further explaining the rationale behind the presented figure design helps to resolve the referee comment.*

**RC1.12:** Line 194: first and only appearance of the acronym SREX, which is not defined
*AR1.12: Thanks, **w**e removed the acronym since it was not necessary.*

**RC1.13:** *Table 2: Awkward spacing. The first and third rows of the table have the text middle-adjusted, while the second and fourth rows have the first column top-adjusted and the other columns bottom-adjusted.*
*AR1.13: The table content has been realigned, thanks for spotting this issue!*

**(4) Overall**

**RC1.14:** The authors have done a moderately good job of addressing the concerns of the initial two referee responses, though a bit more could still be done. Aside from that, this remains a clear, accessible presentation of alternate NDC-based warming scenarios and their impacts.

*AR1.14: Thank you very much for this encouraging feedback! We hope that the presented second round of author responses address the remaining concerns to the referee's satisfaction.*

[revised manuscript text omitted]